

# Kinematic differences between female national and provincial athletes in the tennis serve

Yan Chen[1], Tianyang Wang[1], Yuyan Zhao[2], Genghao Zhan[3], Yinchao Tang[4] and Zefeng Wang[1]

[1] Center for the Research of Sports Psychology and Biomechanics, China Institute of Sport Science, Beijing, Beijing, China
[2] Graduate Department, Capital University of Physical Education and Sports, Beijing, Beijing, China
[3] Department of Fashion and Sports, Capital University of Physical Education and Sports, Beijing, Beijing, China
[4] Sports Training Centre, Hebei Sport University, Hebei, Shijiazhuang, China

Corresponding author
Zefeng Wang, w_tennis@163.com

## ABSTRACT

**Background:** Tennis, the second-largest ball game in the world, has a particularly wide audience. To date, little research has been conducted on the biomechanics of female serves.

**Purpose:** The purpose of this study was to capture the key moments by using 1,000 frames/s high-speed video analysis, to analyze the kinematics of the serving techniques of national athletes and provincial athletes, to determine the differences between the two levels of athletes, and to provide theoretical references for the improvement of scientific training level.

**Methods:** Ten female athletes were selected as participants for this study, five of whom are national athletes, and the other are provincial athletes. Three-dimensional filming techniques were employed to capture videos of the first and second serve techniques. Simi Motion was applied to obtain the 3D kinematic data. Statistical analyses were performed using IBM SPSS Statistics 27.0, and Mann–Whitney U tests were conducted to determine differences between groups.

**Results:** Significant differences in kinematics were found between national and provincial athletes. At the moment of the left knee's minimum flexion (T2), in the first serve, national athletes had a smaller shoulder-hip vertical plane angle ($-18.281 \pm 6.142°$ *vs.* $-25.631 \pm 3.497°$; $p = 0.047$) and a larger hip vertical plane rotation angle ($-9.378 \pm 4.263°$ *vs.* $-0.470 \pm 4.724°$; $p = 0.047$). In the second serve, national athletes had a smaller hip horizontal plane rotation angle ($-1.720 \pm 4.683°$ *vs.* $24.146 \pm 24.014°$; $p = 0.047$) but a larger hip vertical plane rotation angle ($-11.553 \pm 1.949°$ *vs.* $-0.422 \pm 4.958°$; $p = 0.009$). At the moment of impact (T4), in the second serve, national athletes' batting position ($0.296 \pm 0.088$ m *vs.* $0.446 \pm 0.094$ m; $p = 0.047$) was further back. Additionally, in the second serve, national athletes consistently had their body center of gravity further back at T2 ($-0.106 \pm 0.052$ m *vs.* $-0.018 \pm 0.048$ m; $p = 0.028$), T3 ($0.002 \pm 0.038$ m *vs.* $0.132 \pm 0.039$ m; $p = 0.009$), and T4 ($0.073 \pm 0.050$ m *vs.* $0.217 \pm 0.034$ m; $p = 0.009$).

**Conclusions:** The results of this study indicated several significant kinematic differences between national and provincial athletes, these variations were noted in the shoulder, hip, and body center of gravity. In summary, for the overall first and

second serves, it is recommended that national athletes increase the horizontal plane angle of the shoulders and hips at T2, whereas provincial athletes decrease the horizontal plane angle of the shoulder–hip. In addition, provincial athletes need to increase the vertical plane angle of the hip joint, so that the top of the hip can be increased more, and provincial athletes need to be careful not to have the center of gravity too far in front of the body at T2, T3, and T4, so that it can hit the ball at a higher position to increase the swing speed.

# INTRODUCTION

The indispensability of the serve in tennis matches can be seen by the fact that in 2023, the top 20 singles women's tennis players ranked by the Women's Tennis Association (WTA) published tournament data, had a 68 per cent share of first-serve scoring rate in their matches and a 47 per cent share of second-serve scoring rate in their matches. The serve has been described as the most important shot in tennis (*Antúnez et al., 2012*), and is the only shot in one's control and not under the control of the opponent (*Bahamonde, 2000*). A good serve is the beginning of scoring and can give the player an advantage in the match and take the initiative of the game (*Reid, Whiteside & Elliott, 2010*). As the dominant stroke in tennis, it accounts for 45-60% of all strokes in the match (*Johnson et al., 2006*). Because most tournaments consist of three sets of matches, with players averaging 120 serves and 210 ground bats per game, a high-level tennis player can accumulate around 5,400 serves in a competitive season (45 matches per year) (*Myers et al., 2016*).

*Sheets et al. (2011)* have shown that the maximum racket speed occurs within 0.005 s after impact, and while there are no significant differences in the composite racket speed (racket speed at the moment of impact) between different types of serves (flat, kick, and slice serves), there are variations in ball speed. Previous studies on the kinematics of tennis serving techniques have yielded numerous valuable research findings, such as lower leg drive, hip and trunk rotations, and upper arm extension and internal rotations, which seem to be the major contributors to racket and ball speed (*Cohen et al., 1994*; *Colomar et al., 2022*; *Eygendaal, Rahussen & Diercks, 2007*; *Fleisig et al., 2003*; *Genevois et al., 2015*; *Giles & Reid, 2021*; *Gordon & Dapena, 2006*; *Hayes et al., 2021*; *Martin et al., 2013*; *Reid, Elliott & Alderson, 2008*). However, these studies often utilized cameras with recording frequencies ranging from 60 to 500 Hz (*Chow et al., 2003*; *Elliott, Marsh & Blanksby, 1986*; *Giblin, Whiteside & Reid, 2017*; *Giles & Reid, 2021*; *Martin et al., 2013*; *Mitchell, Jones & King, 2010*; *Sheets et al., 2011*; *Vorobiev, Ariel & Dent, 1993*; *Wagner et al., 2014*; *Whiteside et al., 2013a*, *2013b*). Video was recorded using eight synchronized and calibrated VGA cameras (640 × 480 pixels) at 200 Hz (0.005 s per frame interval) (*Sheets et al., 2011*), and *Reid, Campbell & Elliott (2012)* further optimised the shooting conditions; the raw data were collected at 500 Hz (0.005 s per frame interval) with a 12 camera VICON motion analysis system (Vicon, Oxford Metrics Inc, Oxford, United Kingdom), focused on the racket and upper limb ensuring accurate, consistent tracking of the racket and upper limb

at and near impact of the tennis serve. This study employs high-speed cameras (1,000 Hz, 0.005 s per frame interval) to capture videos of players' serving techniques (*Kawazoe & Okimoto, 2009*), which can better capture the key moments and more accurately capture various kinematic data parameters at the key moments, which is conducive to better research.

In addition, much of the research on serve kinematics has focused on male athletes, with very little on female athletes. *Whiteside et al. (2013a)*, in a study on female athletes of different ages (prepubescent: aged 10–11 y, pubescent: aged 14–15 y and postpubescent: aged 18+ y), found that the role of leg drive, shoulder-on-shoulder trunk rotation, internal shoulder rotation, and wrist flexion increases with age, and in turn, older athletes are able to generate higher racket speeds. At the same time, there were no differences in body kinematics between successful serves and service faults, suggesting that service faults cannot be attributed to a single source of biomechanical error (*Whiteside et al., 2013b*).

Based on previous research, the purpose of this article is to capture the key moments through the high-speed video analysis, and then carry out the kinematic analysis of national athletes (23.60 ± 1.00 y) and provincial athletes (17.60 ± 1.50 y), to find out the differences (especially in shoulder-to-shoulder trunk rotation, lower leg drive, shoulder rotation, hip rotation, *etc.*) between the two levels of athletes, and to provide a certain reference for athletes and coaches.

## MATERIALS AND METHODS

### Subject

The movements of the serve of the active national team and Tianjin female athletes were selected as the objects of study. A total of 10 athletes participated in the experiment, of which five of the national team were national athletes, and five of the Tianjin team were provincial athletes, all of which were right-handed (Table 1). According to the questionnaire, it is known that the athletes all use the foot-up serving technique, while the athletes use the slice serve on the first serve and second serve.

National athletes are the title of technical level of Chinese athletes, which is the highest level of sports level title in China, second only to international athletes. In addition, provincial athletes are second only to the national athletes. In addition, national athletes are the top three athletes in singles (singles, doubles, and mixed doubles) and the top two athletes in the team at the National Games. Provincial athletes are the athletes on the court of the 4th to 32nd (including ties) in singles, 4th to 16th (including ties) in doubles (including mixed doubles), and 3rd to 8th in teams at the National Games.

Prior to the experiment, all athletes were in good physical health and condition, enabling them to participate in the tests and perform powerful serves. Participants were informed of the risks involved in performing the experiment and provided informed consent before the formal test. The study design and procedures were in accordance with ethical standards and the Declaration of Helsinki and received approval from the Ethics Committee of the China Institute of Sport Science (20240111).

**Table 1 Athlete basic information.**

|  |  | Age/years | Height/m | Weight/kg | Training years/years |
|---|---|---|---|---|---|
| National | M | 23.60 | 1.70 | 65.00 | 15.20 |
|  | SD | 1.00 | 0.80 | 4.60 | 1.00 |
| Provincial | M | 17.60 | 1.70 | 65.80 | 9.60 |
|  | SD | 1.50 | 0.60 | 2.60 | 1.00 |

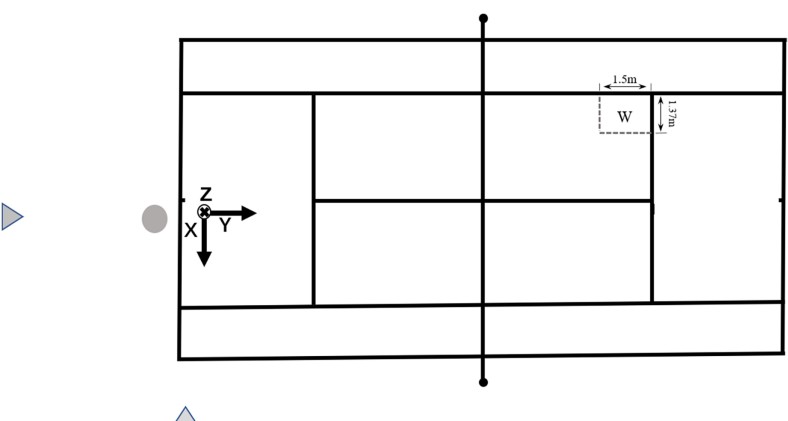

**Figure 1 Schematic diagram of the experimental site.**

## Experimental approach to the problem

At a hard tennis court, in Beihai City on 15 January 2024 and Tianjin City on 20 January 2024, China, from eight o'clock to twelve o'clock, two high-speed cameras (model: Sony DSC-RX10M4, frequency: 1,000 Hz, shutter speed: 1/1,000 s, main optical axis angle ≈ 90°, camera height: 1.5 m) were positioned behind the baseline and 8m to the side of the serving area of the player (Fig. 1).

The camera was calibrated by placing the entire calibration frame in the camera's shooting frame. One of the calibration frames (Peak D20, calibration range: 1.6 × 2.2 × 3.2 m) was mounted on a tripod with a level, which was placed 1 m to the right of the midpoint of the base line for leveling, and the calibration frame covered the subject's torso and limb range of motion.

The delineation of the serving zones employed in previous studies encompassed three distinct configurations: 1 × 1 m$^2$ (*Abrams et al., 2014*; *Reid, Elliott & Alderson, 2007*; *Rosker & Majcen Rosker, 2021*; *Sakurai et al., 2007*), 1.5 × 1.5 m$^2$ (*Martin et al., 2013*), and 2 × 1 m$^2$ (*Reid, Whiteside & Elliott, 2011b*; *Reid et al., 2013*). At the same time, a powerful serve from the outside corner can pull the opponent out of court, leaving the whole court open, so that the whole game is within their control. The outer corners of Zone 1, both for the first and second serves, were defined as the target area measuring 1.37 × 1.5 m$^2$ (Fig. 1).

Prior to the commencement of testing, athletes engaged in a 10-min warm-up session consisting of jogging and serving (*Whiteside et al., 2013b*). Upon attaining a state of preparedness, participants proceeded to the formal testing phase. Athletes were instructed

to execute two serves to the designated target area in the prescribed order of the first serve, followed by a second serve. Sent into the target region and aligned with the line is deemed a valid ball. Among the criteria for a successful serve were that the ball landed within the designated area and that the player had the best feel for the serve. Subsequently, one successful example from each of the first and second serves was selected for further analysis.

### Statistical analyses

The German sports video analysis software Simi Motion 9.1.1 (version number: 9.1.1; SIMI®, Munich, Germany) (*Becker & Russ, 2015*; *Winiarski, 2003*) was used for three-dimensional positioning of the captured video and export of data. The Hanavan body center of mass model was employed for the data analysis (*Hanavan, 1964*). Data processing and analysis were conducted using MATLAB 2016 and Excel 2019 software. IBM SPSS Statistics 27.0 (Chicago, IL, USA). Mann–Whitney U tests were conducted to determine differences between groups. Statistical significance was set at $p = 0.05$. In this study, the data results are expressed as mean ± standard deviation (M ± SD) according to the difference relationship.

## CONCEPT DEFINITIONS

### Court and racket definitions

The origin of the two court coordinate systems is set at the midpoint of the baseline. The X-axis positive direction (left to right) is defined as pointing to the right and parallel to the baseline. The Y-axis positive direction (front to back) is defined as pointing forward and parallel to the single sideline. The Z-axis positive direction (up and down) is defined as pointing upward from the midpoint of the baseline. Furthermore, the racket's rotational state at the moment of impact is described using Euler angles (Z-Y-X). The midpoint of the racket refers to the center point of the line connecting the left and right sides of the racket. The Z-axis direction of the racket points upward towards the racket head, the Y-axis direction points forward and is perpendicular to the racket surface, and the X-axis direction points to the right and is perpendicular to the midpoint of the racket. In three-dimensional space, the Z-axis, Y-axis, and X-axis are mutually perpendicular. The positive and negative values of the data only represent the direction and do not indicate magnitude (Fig. 2). The state of the racket is shown in Fig. 3.

### Divisions of key moments of the serve

Previous researchers have divided the technical action of serving into four moments: ball release, trophy position, racket low point, and impact (*Whiteside et al., 2013a*, *2013b*). This article refers to previous research and divides the serving action into four moments: the moment the ball leaves the hand (T1), the moment of the left knee's minimum flexion (T2), the moment of the lowest point of the racket head (T3), and the moment of impact (T4). See Fig. 4 for details of the key moments of the serve.

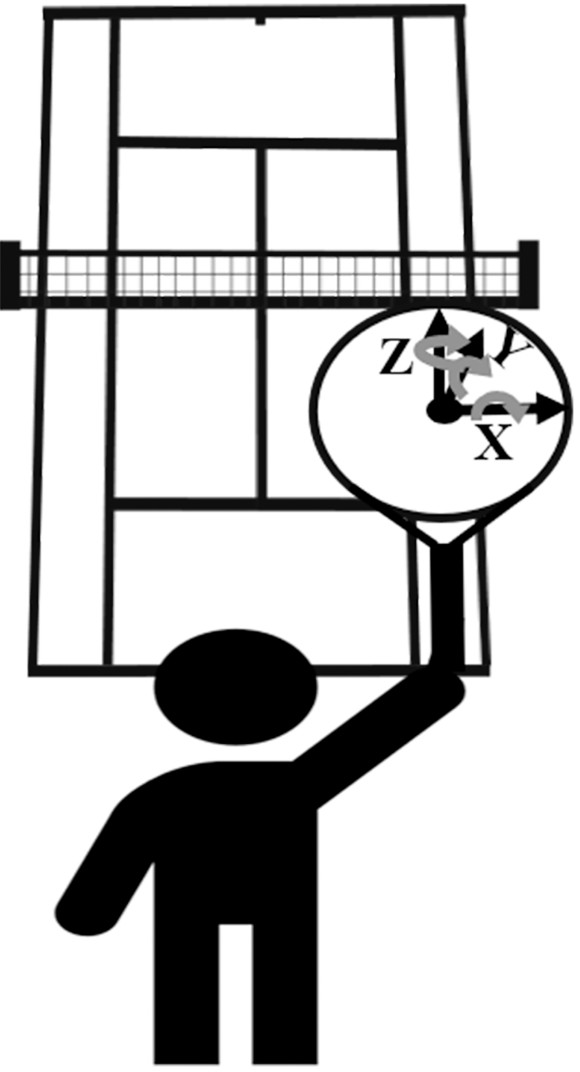

**Figure 2  Athlete facing the court diagram.**     

## RESULTS

### Comparison of differences between first and second serves

#### *In terms of ball speed*

As shown in Table 2, there was no significant difference between national and provincial athletes in the first and second serve speed and swing speed (swing speed is the speed of the racket before hitting the ball.). National athletes were faster in the first serve than provincial athletes (145.234 ± 10.224 km/h *vs.* 142.358 ± 11.090 km/h; $p$ = 0.754) and slower in the second serve than provincial athletes (128.705 ± 2.713 km/h *vs.* 130.496 ± 10.603 km/h; $p$ = 0.347). In terms of swing speed, national athletes were faster than provincial athletes (146.354 ± 15.181 km/h *vs.* 145.550 ± 18.274 km/h; $p$ = 0.917) in the first serves and the second serves were slower than provincial athletes (140.635 ± 12.083 km/h *vs.* 146.108 ± 18.559 km/h; $p$ = 0.465).

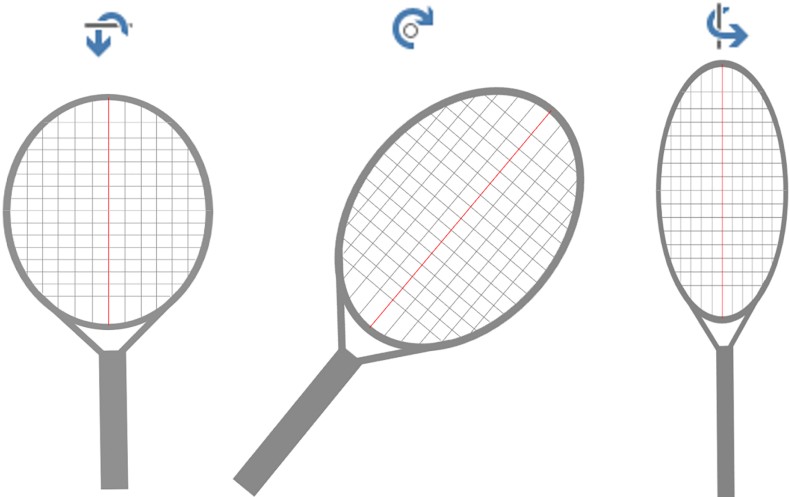

**Figure 3 Racket status diagram.** From left to right: press downward, tilt to the right and rotate outward. The opposite direction: tilt upward, tilt to the left, and rotate inward.

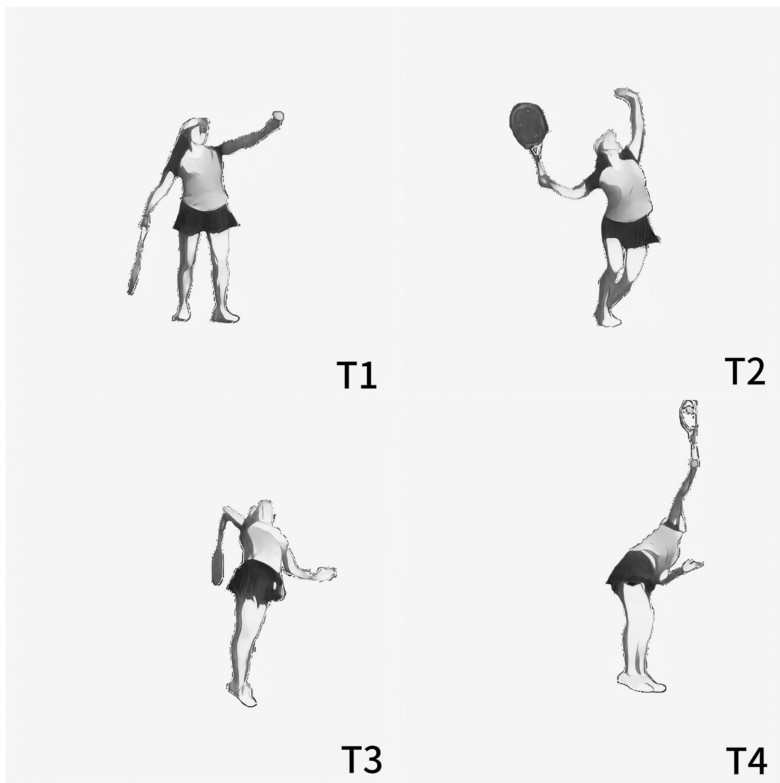

**Figure 4 Key moments of the serve.** T1, the moment the ball leaves the hand; T2, the moment of the left knee's minimum flexion; T3, the moment of the lowest point of the racket head; T4, the moment of impact.

**Table 2 Comparison of the first and second serve and swing speed between female national and provincial athletes.**

| Variable | | First serve | | p | Z | Second serve | | p | Z |
|---|---|---|---|---|---|---|---|---|---|
| | | National M ± SD | Provincial M ± SD | | | National M ± SD | Provincial M ± SD | | |
| Ball speed (km/h) | VTotal | 145.234 ± 10.224 | 142.358 ± 11.090 | 0.754 | −0.313 | 128.705 ± 2.713 | 130.496 ± 10.603 | 0.347 | −0.94 |
| | Vx | −31.314 ± 4.714 | −33.292 ± 7.439 | 0.754 | −0.313 | −24.437 ± 2.903 | −28.788 ± 4.851 | 0.175 | −1.358 |
| | Vy | 141.491 ± 9.720 | 137.933 ± 9.238 | 0.602 | −0.522 | 126.274 ± 2.543 | 127.011 ± 10.048 | 0.465 | −0.731 |
| | Vz | −8.889 ± 2.215 | −9.819 ± 5.181 | 0.754 | −0.313 | −3.644 ± 2.041 | −6.867 ± 3.871 | 0.251 | −1.149 |
| Swing speed (km/h) | VTotal | 146.354 ± 15.181 | 145.550 ± 18.274 | 0.917 | −0.104 | 140.635 ± 12.083 | 146.108 ± 18.559 | 0.465 | −0.731 |
| | Vx | 11.818 ± 21.505 | 2.377 ± 10.781 | 0.602 | −0.522 | 19.469 ± 16.343 | 16.029 ± 10.728 | 0.917 | −0.104 |
| | Vy | 142.504 ± 16.208 | 144.422 ± 17.522 | 0.917 | −0.104 | 134.030 ± 10.983 | 141.884 ± 16.927 | 0.347 | −0.94 |
| | Vz | 22.532 ± 9.285 | 13.622 ± 8.964 | 0.175 | −1.358 | 34.255 ± 9.250 | 29.132 ± 9.025 | 0.347 | −0.94 |

**Table 3 Comparison of first and second serve kinematic parameters between female national and provincial athletes at T1.**

| Variable | | First serve | | p | Z | Second serve | | p | Z |
|---|---|---|---|---|---|---|---|---|---|
| | | National M ± SD | Provincial M ± SD | | | National M ± SD | Provincial M ± SD | | |
| Ball (m) | X | 1.310 ± 0.334 | 1.166 ± 0.270 | 0.750 | −0.313 | 1.320 ± 0.334 | 1.153 ± 0.245 | 0.602 | −0.522 |
| | Y | 0.095 ± 0.147 | 0.164 ± 0.060 | 0.175 | −1.358 | 0.082 ± 0.139 | 0.160 ± 0.071 | 0.465 | −0.731 |
| | Z | 1.640 ± 0.132 | 1.589 ± 0.157 | 0.754 | −0.313 | 1.645 ± 0.109 | 1.584 ± 0.164 | 0.754 | −0.313 |
| Shoulder-hip angle (°) | Horizontal plane | 0.124 ± 2.052 | 5.101 ± 2.612 | 0.016* | −2.402 | −0.550 ± 6.349 | 5.412 ± 3.702 | 0.117 | −1.567 |
| | Vertical plane | −3.678 ± 4.512 | −5.627 ± 3.346 | 0.602 | −0.522 | −6.029 ± 3.820 | −7.201 ± 7.195 | 0.917 | −0.104 |
| Body angle (°) | Left knee | 165.952 ± 7.246 | 168.982 ± 6.723 | 0.465 | −0.731 | 163.833 ± 9.748 | 170.266 ± 5.037 | 0.347 | −0.940 |
| | Right knee | 166.484 ± 4.926 | 160.879 ± 10.944 | 0.602 | −0.522 | 166.917 ± 6.751 | 156.300 ± 13.340 | 0.175 | −1.358 |
| | Left ankle | 105.829 ± 8.916 | 106.543 ± 9.151 | 0.917 | −0.104 | 106.626 ± 12.288 | 104.443 ± 5.544 | 0.917 | −0.104 |
| | Right ankle | 95.401 ± 7.177 | 88.666 ± 6.776 | 0.175 | −1.358 | 99.650 ± 5.621 | 88.288 ± 5.936 | 0.016* | −2.402 |
| Center of gravity (m) | X | 0.754 ± 0.343 | 0.739 ± 0.276 | 0.917 | −0.104 | 0.755 ± 0.350 | 0.718 ± 0.287 | 0.917 | −0.104 |
| | Y | −0.305 ± 0.083 | −0.408 ± 0.110 | 0.117 | −1.567 | −0.300 ± 0.076 | −0.412 ± 0.112 | 0.076 | −1.776 |
| | Z | 1.011 ± 0.080 | 1.030 ± 0.081 | 0.754 | −0.313 | 1.005 ± 0.068 | 1.021 ± 0.067 | 0.754 | −0.313 |

Note:
* $p < 0.05$.

### At the moment the ball leaves the hand (T1)

As shown in Table 3, in the first serve, there were significant differences in the shoulder-hip horizontal plane angle and right elbow, the shoulder-hip horizontal plane angle (0.124 ± 2.052° vs. 5.101 ± 2.612°; $p = 0.016$) and the right elbow angle (140.784 ± 37.762° vs. 173.033 ± 3.670°; $p = 0.028$) were smaller in national athletes than in provincial athletes. In the second serve, there was a significant difference in the right ankle angle, and the right ankle extension was greater in national athletes (99.650° vs. 88.288°; $P = 0.016$).

### At the moment of the left knee's minimum flexion (T2)

As shown in Table 4, in the first serve, there was a significant difference in the vertical plane of the shoulder-hip angle, the angle in national athletes was less than that in

**Table 4 Comparison of first and second serve kinematic parameters between female national and provincial athletes at T2.**

| Variable | | First serve | | p | Z | Second serve | | p | Z |
|---|---|---|---|---|---|---|---|---|---|
| | | National M ± SD | Provincial M ± SD | | | National M ± SD | Provincial M ± SD | | |
| Shoulder-hip angle (°) | Horizontal plane | −9.762 ± 10.852 | −37.890 ± 34.989 | 0.347 | −0.940 | −9.004 ± 8.837 | -19.894 ± 30.858 | 0.917 | −0.104 |
| | Vertical plane | −18.281 ± 6.142 | −25.631 ± 3.497 | 0.047* | −1.984 | −19.604 ± 10.761 | -25.493 ± 8.232 | 0.602 | −0.522 |
| Shoulder rotation angle (°) | Horizontal plane | −8.092 ± 15.512 | −7.038 ± 20.861 | 0.917 | −0.104 | −10.178 ± 13.214 | -1.161 ± 11.537 | 0.175 | −1.358 |
| | Vertical plane | −23.981 ± 11.059 | −20.473 ± 5.914 | 0.465 | −0.731 | −25.129 ± 11.195 | -18.713 ± 4.683 | 0.175 | −1.358 |
| Hip rotation angle (°) | Horizontal plane | 1.793 ± 8.457 | 35.953 ± 20.817 | 0.076 | −1.776 | −1.720 ± 4.683 | 24.146 ± 24.014 | 0.047* | −1.984 |
| | Vertical plane | −9.378 ± 4.263 | −0.470 ± 4.724 | 0.047* | −1.984 | −11.553 ± 1.949 | -0.422 ± 4.958 | 0.009* | −2.611 |
| Body angle (°) | Left knee | 115.252 ± 12.188 | 114.300 ± 7.212 | 0.754 | −0.313 | 120.662 ± 12.957 | 117.597 ± 9.076 | 0.602 | −0.522 |
| | Right knee | 108.472 ± 3.763 | 107.822 ± 710.103 | 0.465 | −0.731 | 111.178 ± 6.246 | 101.155 ± 10.849 | 0.347 | −0.940 |
| | Left ankle | 82.429 ± 6.943 | 78.690 ± 6.047 | 0.465 | −0.731 | 90.205 ± 2.414 | 78.755 ± 5.522 | 0.009* | −2.611 |
| | Right ankle | 88.292 ± 8.457 | 92.656 ± 8.053 | 0.347 | −0.940 | 86.014 ± 9.910 | 90.567 ± 5.977 | 0.754 | −0.313 |
| Center of gravity (m) | X | 0.882 ± 0.308 | 0.915 ± 0.270 | 0.754 | −0.313 | 0.887 ± 0.316 | 0.903 ± 0.266 | 0.917 | −0.104 |
| | Y | −0.086 ± 0.067 | −0.028 ± 0.042 | 0.117 | −1.567 | −0.106 ± 0.052 | -0.018 ± 0.048 | 0.028* | −2.193 |
| | Z | 0.893 ± 0.074 | 0.915 ± 0.062 | 0.602 | −0.522 | 0.899 ± 0.068 | 0.905 ± 0.068 | 0.917 | −0.104 |

Note:
* $p < 0.05$.

provincial athletes (−18.281 ± 6.142° vs. −25.631 ± 3.497°; $p = 0.047$). There was a significant difference in the shoulder-hip vertical plane angle, and the national athletes hip rotated more downward (−9.378 ± 4.263° vs. −0.470 ± 4.724°; $p = 0.047$; $p = 0.047$). In the second serve, there were significant differences in the horizontal and vertical planes of hip rotation, with the national athletes hip rotating backward and the provincial athletes hip rotating forward (−1.720 ± 4.683° vs. 24.146 ± 24.014°; $p = 0.047$), and the hip tilted downward, with a greater degree of tilt (−11.553 ± 1.949° vs. −0.422 ± 4.958°; $p = 0.009$). Left ankle angle extension was greater in national athletes (90.21° vs. 78.75°; $p = 0.009$) and the center of gravity was pressed back more (−0.106 ± 0.052 m vs. −0.018 ± 0.048 m; $p = 0.028$).

### At the moment of the lowest point of the racket head (T3)

As shown in Table 5, in the first serve, there was a significant difference in the horizontal plane of the shoulder-hip angle, the national athletes shoulder was turned forward to the hip, and provincial athletes turned shoulder-backward to the hip (0.428 ± 6.423° vs. −9.67 ± 7.260°; $p = 0.047$).

There were significant differences in the vertical plane of shoulder (−31.044 ± 8.188° vs. −49.913 ± 4.731°; $p = 0.009$) and hip (−19.057 ± 5.113° vs. −27.897 ± 5.735°; $p = 0.028$) rotation, and the rotation amplitude of national athletes was smaller than that of provincial athletes. In addition, the right knee angle (156.544 ± 13.054° vs. 173.233 ± 3.058°;

**Table 5 Comparison of first and second serve kinematic parameters between female national and provincial athletes at T3.**

| Variable | | First serve | | p | Z | Second serve | | p | Z |
|---|---|---|---|---|---|---|---|---|---|
| | | National M ± SD | Provincial M ± SD | | | National M ± SD | Provincial M ± SD | | |
| Shoulder-hip angle (°) | horizontal plane | 0.428 ± 6.423 | −9.670 ± 7.260 | 0.047* | −1.984 | −3.524 ± 11.363 | −14.924 ± 5.060 | 0.117 | −1.567 |
| | vertical plane | −6.294 ± 3.736 | −3.616 ± 5.699 | 0.465 | −0.731 | −6.295 ± 4.471 | −5.725 ± 3.730 | 0.917 | −0.104 |
| Shoulder rotation angle (°) | horizontal plane | 30.674 ± 8.511 | 54.013 ± 22.280 | 0.175 | −1.358 | 30.195 ± 5.825 | 46.134 ± 17.301 | 0.076 | −1.776 |
| | vertical plane | −31.044 ± 8.188 | −49.913 ± 4.731 | 0.009* | −2.611 | −34.660 ± 11.956 | −51.751 ± 12.461 | 0.047* | −1.984 |
| Hip rotation angle (°) | horizontal plane | 20.484 ± 8.928 | 25.792 ± 20.865 | 0.917 | −0.104 | 24.715 ± 20.463 | 41.163 ± 19.654 | 0.175 | −1.358 |
| | vertical plane | −19.057 ± 5.113 | −27.897 ± 5.735 | 0.028* | −2.193 | −21.351 ± 5.587 | −31.983 ± 6.596 | 0.047* | −1.984 |
| Body angle (°) | Left knee | 173.832 ± 5.114 | 171.703 ± 4.810 | 0.347 | −0.940 | 170.485 ± 6.584 | 169.753 ± 5.932 | 0.754 | −0.313 |
| | Right knee | 156.544 ± 13.054 | 173.233 ± 3.058 | 0.009* | −2.611 | 167.063 ± 7.515 | 172.799 ± 4.400 | 0.251 | −1.149 |
| | Left ankle | 137.585 ± 8.832 | 128.018 ± 15.314 | 0.175 | −1.358 | 137.312 ± 9.500 | 126.725 ± 10.794 | 0.465 | −0.731 |
| | Right ankle | 143.417 ± 3.987 | 142.265 ± 8.885 | 0.754 | −0.313 | 146.671 ± 7.923 | 143.762 ± 6.120 | 0.465 | −0.731 |
| Center of gravity (m) | X | 0.862 ± 0.322 | 0.899 ± 0.254 | 0.917 | −0.104 | 0.873 ± 0.316 | 0.867 ± 0.265 | 0.917 | −0.104 |
| | Y | 0.032 ± 0.063 | 0.134 ± 0.057 | 0.028* | −2.193 | 0.002 ± 0.038 | 0.132 ± 0.039 | 0.009* | −2.611 |
| | Z | 1.085 ± 0.059 | 1.114 ± 0.071 | 0.465 | −0.731 | 1.077 ± 0.051 | 1.097 ± 0.076 | 0.754 | −0.313 |

**Note:**
* $p < 0.05$.

$p = 0.009$) and body center of gravity Y (0.03 *vs.* 0.13 m; $p = 0.028$) also differed, with the right knee angle and the anterior center of gravity of the national athletes was smaller than the provincial athletes. In the second serve, significant differences were observed in the vertical plane of the shoulder (−34.660 ± 11.956° *vs.* −51.751 ± 12.461°; $p = 0.047$), hip rotation (−21.351 ± 5.587° *vs.* −31.983 ± 6.596°; $p = 0.047$), and body center of gravity Y (0.002 ± 0.038 m *vs.* 0.132 ± 0.039 m; $p = 0.009$).

### At the moment of impact (T4)

As shown in Table 6, in terms of the first serve, there was a significant difference in the left ankle angle (133.721 ± 9.858° *vs.* 112.782 ± 16.230°; $p = 0.047$). In terms of the second serve, there were also significant differences in the left hip angle, left ankle angle, and center of gravity Y. National athletes had better extension of the left ankle angle (140.803 ± 5.363° *vs.* 125.279 ± 8.232°; $p = 0.009$), left ankle angle (138.898 ± 9.447° *vs.* 110.429 ± 8.334°; $p = 0.009$), and a more backward center of gravity (0.073 ± 0.050 m *vs.* 0.217 ± 0.034 m; $p = 0.009$). There was a significant difference in batting position Y, with national athletes hitting position being more backward (0.296 ± 0.088 m *vs.* 0.446 ± 0.094 m; $p = 0.047$).

## DISCUSSION

The results of this study indicated several significant kinematic differences between national and provincial athletes. These variations were noted in the shoulder rotation, hip rotation, and body center of gravity.

**Table 6 Comparison of first and second serve kinematic parameters between female national and provincial athletes at T4.**

| Variable | | First serve | | *p* | Z | Second serve | | *p* | Z |
|---|---|---|---|---|---|---|---|---|---|
| | | National M ± SD | Provincial M ± SD | | | National M ± SD | Provincial M ± SD | | |
| Ball (m) | X | 0.862 ± 0.522 | 0.933 ± 0.327 | 0.917 | −0.104 | 0.771 ± 0.400 | 0.852 ± 0.299 | 0.602 | −0.522 |
| | Y | 0.373 ± 0.153 | 0.489 ± 0.147 | 0.175 | −1.358 | 0.296 ± 0.088 | 0.446 ± 0.094 | 0.047* | −1.984 |
| | Z | 2.608 ± 0.099 | 2.612 ± 0.115 | 0.917 | −0.104 | 2.626 ± 0.116 | 2.616 ± 0.148 | 0.917 | −0.104 |
| Shoulder-hip angle (°) | Horizontal plane | 14.818 ± 13.501 | 21.709 ± 9.431 | 0.251 | −1.149 | 14.741 ± 20.015 | 17.924 ± 9.866 | 0.917 | −0.104 |
| | Vertical plane | 34.614 ± 12.792 | 35.408 ± 8.084 | 0.917 | −0.104 | 35.419 ± 17.810 | 35.586 ± 5.701 | 0.754 | −0.313 |
| Shoulder rotation angle (°) | Horizontal plane | 55.600 ± 32.926 | 49.128 ± 14.128 | 0.465 | −0.731 | 42.945 ± 34.300 | 48.691 ± 9.983 | 0.754 | −0.313 |
| | Vertical plane | 86.143 ± 24.495 | 63.034 ± 10.865 | 0.251 | −1.149 | 76.128 ± 25.437 | 61.181 ± 6.148 | 0.465 | −0.731 |
| Hip rotation angle (°) | Horizontal plane | 41.209 ± 27.779 | 17.749 ± 10.074 | 0.251 | −1.149 | 24.680 ± 17.480 | 15.843 ± 5.872 | 0.465 | −0.731 |
| | Vertical plane | 45.235 ± 33.844 | 24.010 ± 6.564 | 0.602 | −0.522 | 34.414 ± 31.170 | 19.870 ± 5.591 | 0.917 | −0.104 |
| Body angle (°) | Left knee | 168.467 ± 6.106 | 164.158 ± 9.607 | 0.251 | −1.149 | 170.164 ± 3.242 | 173.933 ± 3.356 | 0.076 | −1.776 |
| | Right knee | 170.900 ± 2.986 | 170.903 ± 6.041 | 0.602 | −0.522 | 172.640 ± 2.208 | 170.731 ± 6.938 | 0.754 | −0.313 |
| | Left ankle | 133.721 ± 9.858 | 112.782 ± 16.230 | 0.047* | −1.984 | 138.898 ± 9.447 | 110.429 ± 8.334 | 0.009* | −2.611 |
| | Right ankle | 142.131 ± 7.676 | 128.136 ± 13.096 | 0.175 | −1.358 | 140.279 ± 5.550 | 131.616 ± 14.057 | 0.251 | −1.149 |
| Center of gravity (m) | X | 0.840 ± 0.318 | 0.892 ± 0.259 | 0.754 | −0.313 | 0.854 ± 0.313 | 0.844 ± 0.266 | 0.917 | −0.104 |
| | Y | 0.102 ± 0.052 | 0.218 ± 0.084 | 0.076 | −1.776 | 0.073 ± 0.050 | 0.217 ± 0.034 | 0.009* | −2.611 |
| | Z | 1.190 ± 0.058 | 1.175 ± 0.072 | 0.602 | −0.522 | 1.181 ± 0.048 | 1.168 ± 0.082 | 0.465 | −0.731 |

**Note:**
* $p < 0.05$.

## The speed of the racket and ball

There was a correlation between the speed of the racket and the speed of the ball at T4 (*Gordon & Dapena, 2006*; *Tanabe & Ito, 2007*), which contributed positively to overall service performance (*Fleisig et al., 2003*; *Girard, Micallef & Millet, 2007*). It could also be found that the swing speed of the racket has determined the speed of the ball to a certain extent. Usually there will be more topspin on the second serve, and more "brushing" when the dribble hits (*Abrams et al., 2014*). This study demonstrated that the athletes featured in this article exhibit slower ball speeds in their first serves compared to the findings of *Whiteside et al.'s (2013a)* previous research (national athletes, provincial athletes: 145.234, 142.358 km/h *vs*. postpubescent: 156.24 km/h), it is possible that this discrepancy can be attributed to differences in serving styles.

*Vorobiev, Ariel & Dent (1993)* studied Andre Agassi's serve and investigated the biomechanical differences between the flat (first) serve and kick (second) serve. They found that for this specific high-level player, the horizontal center of gravity velocity was greater in the flat serve, whereas the kick serve demonstrated a larger vertical center of gravity linear velocity. This investigation also showed that the kick has a more medial (over the head) contact point than the flat serve (*Vorobiev, Ariel & Dent, 1993*). The data on ball position and swing speed on the first and second serves showed that national athletes and

provincial athletes the ball more by increasing the vertical and lateral racket speeds and decreasing the forward speeds from the first to the second serve (Table 2), throwing the ball closer to the body and imparting topspin and sidespin on the ball (Tables 3 and 6), which led to a decrease in the overall ball speed.

### In terms of the ball

The stability of the tennis ball throwing action is crucial for the player's later strokes, and the height of the ball throw and the trajectory of the ball will affect the accuracy and success of the serve (*Brody, 1987*; *Reid, Whiteside & Elliott, 2011a*). The reasonable throwing technique is to properly adjust the elbow joint of the throwing arm and control the movement of the wrist joint during the throwing process, so as to ensure that the ball is thrown vertically. Studies have shown that from the first to the second serve, the player throws the ball closer to the body, resulting in a decrease in the forward position of the ball before hitting and an increase in the vertical and lateral velocities of the racket before T4, resulting in kick and slice spin (*Chow et al., 2003*; *Gordon & Dapena, 2006*).

According to this study, both the first serves of national athletes and provincial athletes were positioned further forward than their second serves, which aligns with prior research indicating that first serves are typically thrown more forward. Furthermore, the research revealed that the first serve batting position (0.373 ± 0.153, 0.489 ± 0.147 m) for national athletes and provincial athletes was more advanced than the second serve batting position (0.296 ± 0.088, 0.446 ± 0.094 m), According to the data, it appears that the variation in batting distance between the initial and subsequent serves of the national athletes (0.077 m) is larger than that of the provincial athletes (0.043 m), which may explain the disparity in the speed of their second serves.

At the same time, through the data, it can be seen that on the X-axis, the ball flies from right to left from T1 to T4, and on the Y-axis, the ball flies from the front (away from the body) to the back (close to the body), and does not move straight up and down. This is consistent with the finding of *Chow et al. (2003)* that during the flight phase of the toss, the ball moved forward and to the left on all.

### In terms of shoulder-hip angle

The approximate order of the main contributors to racket speed between T2 and T4: external rotation of the shoulder, wrist extension, torsional rotation of the lower torso, torsional rotation of the upper trunk relative to the lower trunk, shoulder abduction, elbow extension, ulnar deflection rotation, second torsional rotation of the upper trunk relative to the lower trunk, and wrist flexion. Among them, elbow extension and wrist flexion contribute particularly well (*Gordon & Dapena, 2006*). Among them, the relevant rotation corresponds to the angle of shoulder rotation, hip rotation angle and shoulder-hip angle in this study. There have been reported that the lateral rotation of the torso makes a moderate contribution to the batting speed of the racket (*Elliott, Marshall & Noffal, 1995*). It has also been proposed that high-level men's tennis serves to transfer momentum to the power arm through lateral (torsion), frontal (shoulder-to-shoulder) and sagittal plane trunk rotation (*Bahamonde, 2000*).

Previous results showed that the contact with the ground during the tennis serve caused the front leg to have greater rotation, and then the trunk and pelvis rotation accelerated longer, so that the pelvic maximum angular velocity and the trunk maximum angular velocity were produced during the tennis serve, and the maximum angular velocity of the tennis serve would be higher (*Wagner et al., 2014*), so the importance of the horizontal angle between the shoulder and hip can be seen. There are also studies that show that at the end of the buffer (the moment of the left knee's minimum flexion), the angle between the shoulders and the hips is 20–30° (*Elliott, Reid & Crespo, 2003*), the disparity between the shoulder-hip angle of athletes in both groups as presented in this article is noteworthy, and it is essential for athletes to be cognizant of this characteristic.

In addition, this study also found that there was a significant difference in the vertical surface of the shoulder-hip angle at T2 in the first serve, and the national athletes were smaller than those of provincial athletes, which meant that the amplitude of shoulder and hip movement of provincial athletes was greater. At T3, there was a significant difference in the horizontal plane of the shoulder-hip angle, with national athletes turning the shoulder forward towards the hip and provincial athletes turning the shoulder backward towards the hip.

## In terms of shoulder and hip rotation angles

There were significant differences in the vertical rotation angles of the first and second serves at T2, which was mainly manifested in the fact that the national athletes' hip tilts more downward, which may be the result of the need for a hip top. At T3, there was a significant difference in the rotation amplitude of the vertical plane of the shoulder, and the rotation angle of national athletes was smaller than that of provincial athletes. At the same time, it can be observed that there was a difference in the rotation angle of the vertical plane of the first and second serves, and the national athletes' rotation angle was smaller than that of provincial athletes. The torso horizontal rotation from T2 to T4 increases the distance from the racket to the striking position and the speed of the racket at T4. In addition, these rotations pre-stretch the trunk muscles and store elastic energy (*Tubez et al., 2021*). The angle of rotation of the shoulder and hip horizons of national athletes from T2 to T4 was greater than that of provincial athletes, indicating that national athletes had a better grasp of this aspect when serving.

## In terms of body angles

Leg drive played a crucial precursor role in generating powerful torso rotation (*Whiteside et al., 2013a*). The knee and hip velocities generated by the lower limb drive were synchronized with the shoulder and elbow velocities generated by the torso and upper arm movements, followed by elbow extension and finally forearm internal rotation and wrist flexion (*Elliott & Marsh, 1989*). Studies have shown that knee flexion helps to generate more power and speed when the leg is driven (*Bonnefoy et al., 2009*; *Girard, Micallef & Millet, 2005*; *Hornestam et al., 2021*). In this study, the left and right knee angles of national athletes were larger than those of provincial athletes at T2. In addition, elbow extension and wrist flexion were particularly important contributors to racket speed (*Gordon &*

*Dapena, 2006*), and the change in national athletes right elbow angle from T3 to T4 was greater than that of provincial athletes in this study. *Whiteside et al. (2013a)* found that the ankle extension was greater when the leg was driven during the serve than the knee or hip. Plantar flexors were major contributors to the forward propulsion of gait (*Elliott & Reid, 2004*), and this muscle group played a crucial role in leg drive in women. When the foot was fixed in the trophy position on the ground, knee flexion prior to leg drive was essential for placing the plantar flexors (*Elliott & Reid, 2004*). There was a significant difference in the left ankle angle at the moment of the first and second serves, with the athlete having a larger left ankle angle, which also meant that national athletes' legs were more fully extended, and their bodies were better stretched when batting the ball.

### In terms of body center of gravity

From the moment of minimal left knee flexion to the moment of impact, the provincial athletes body center of gravity is more forward, which leads to a more forward strike position. *Dossena et al. (2018)* worked out that reaching higher impact point during tennis serve could allow to serve faster. Through Z, it was found that the center of gravity of the national athletes was lower than that of provincial athletes at T1, T2, and T3, but was higher than that of provincial athletes at T4, which showed that national athletes had always maintained a low center of gravity before hitting the ball, obtained upward energy through the rapid push and extension of the knee joint, and then hit the ball at a high position to obtain faster ball speed and spin.

### Study limitations

This study not only found significant differences between the two groups of athletes in terms of shoulder-hip and shoulder-rotation and hip-rotation angles, but also found significant difference in body center of gravity. However, a small sample size is considered a limitation of this study, future studies should focus on larger samples of participants. Other factors that may affect the effectiveness of the serve (psychological stress, external environmental disturbances) are not added to the experiment, and it is expected that these disturbances will be added in future studies. In addition, intervention studies can be included to observe differences in athletes and explore training pathways for technique optimization.

## CONCLUSIONS

The results of this study indicated several significant kinematic differences between national and provincial athletes, these variations were noted in the shoulder, hip, and body center of gravity. In summary, for the overall first and second serves, it is recommended that national athletes increase the horizontal plane angle of the shoulders and hips at T2, whereas provincial athletes decrease the horizontal plane angle of the shoulder–hip. In addition, provincial athletes need to increase the vertical plane angle of the hip joint, so that the top of the hip can be increased more, and provincial athletes need to be careful not to have the center of gravity too far in front of the body at T2, T3, and T4, so that it can hit the ball at a higher position, which in turn can increase the swing speed. Based on the

results of these analyses, coaches need to recognize the importance of key metrics and continually adjust training plans based on these metric differences.

### Funding

This work was supported by the China Institute of Sport Science (Project 22-37 Supported by the Fundamental Research Funds for the China Institute of Sport Science). The funders had no role in study design, data collection and analysis, decision to publish, or preparation of the manuscript.

### Grant Disclosures

The following grant information was disclosed by the authors:
China Institute of Sport Science: Project 22–37.

### Competing Interests

The authors declare that they have no competing interests.

### Author Contributions

- Yan Chen conceived and designed the experiments, performed the experiments, analyzed the data, prepared figures and/or tables, authored or reviewed drafts of the article, and approved the final draft.
- Tianyang Wang conceived and designed the experiments, performed the experiments, analyzed the data, prepared figures and/or tables, authored or reviewed drafts of the article, and approved the final draft.
- Yuyan Zhao performed the experiments, analyzed the data, authored or reviewed drafts of the article, and approved the final draft.
- Genghao Zhan performed the experiments, authored or reviewed drafts of the article, and approved the final draft.
- Yinchao Tang performed the experiments, authored or reviewed drafts of the article, and approved the final draft.
- Zefeng Wang conceived and designed the experiments, performed the experiments, prepared figures and/or tables, authored or reviewed drafts of the article, and approved the final draft.

### Human Ethics

The following information was supplied relating to ethical approvals (*i.e.*, approving body and any reference numbers):

The China Institute of Sport Science granted Ethical approval to carry out the study within its facilities (20240111).

### Data Availability

The raw measurements are available in the Supplemental File.

## Supplemental Information

Supplemental information for this article can be found online at http://dx.doi.org/10.7717/peerj.18410#supplemental-information.

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
