# Peer review of "Kinematic differences between female national and provincial athletes in the tennis serve"

_PeerJ, doi:10.7717/peerj.18410_

## Round 0.1 · original submission · Major Revisions

Reviewers expressed concerns that the introduction did not set up the research question. The small sample size was also a concern for the between subject analysis. Reviewers also suggested that this manuscript should be edited by a native English speaker. I have decided to allow the authors the opportunity to respond to these comments with the understanding that major revisions are required.

·

Basic reporting

see attached file

Experimental design

see attached file

Validity of the findings

see attached file

Additional comments

see attached file

Reviewer 2 ·

Basic reporting

The manuscript should be edited by a primary English speaker.

References: OK

Introduction: In general, it was not written in accordance with the purpose of the study. It is necessary to provide literature information about why kinematic differences may occur during serve in tennis players at different athletic levels.

Tables: Table 2 and Table 3 are overloaded. Kinematic data that do not come to the fore during the serve can be deleted from the tables.

Figures: Drawing quality is very low. In particular, the drawing quality of image 4 needs to be improved. May be real images of the athletes should be added.

Experimental design

The study meets the scope of the journal
Research question: OK
Ethical standard: OK
Methods section needs to be improved.
Line 110: Why a separate target area specified in the service box? Were the serves hit below side of the W square (the most effective area) deemed to be invalid?
Line 114: Did all athletes perform the same type of serve? Please insert the type (flat, spin, or kick) of the serve.
Line 121: “The Hanavan body center of mass model was employed for data analysis.” Please insert a reference.
Line 143 and Figure 4: Phases of the serve should be named as; beginning of the preparation phase (T1), beginning of the acceleration phase (T2), end of the acceleration phase (T3), and beginning of the follow-through phase (T4).
Related references
Deniz V, Sariyildiz A, Buyuktas B, Basaran S. Comparison of the activation and mechanical properties of scapulothoracic muscles in young tennis players with and without scapular dyskinesis: an observational comparative study. J Shoulder Elbow Surg. 2024 Jan;33(1):192-201. doi: 10.1016/j.jse.2023.07.016.
Kovacs M, Ellenbecker T. An 8-stage model for evaluating the tenis serve: implications for performance enhancement and injury prevention.
Sports Health 2011;3:504-13. https://doi.org/10.1177/1941738111414175.

Validity of the findings

Results: OK
Please summarize in the first paragraph of the discussion section any results that you think may be important to readers.

Line 191: “According to the data, the MS first serve speed (swing speed) is higher than DIA…” Since there is no statistically significant difference, no expression should be given in this regard.

Why was there no significant difference in ball speed even though there was a difference in kinematic data between the two groups? I think this should be the most important question to focus on in this section?

What is/are the limitation(s) of the study? Study sample size may be one of the limitations.

Conclusions: OK

Additional comments

Thank you for the opportunity to read and review the manuscript entitled: Kinematic differences in serving techniques between female tennis master sportsman and division I athletes. Determining the possible kinematic changes that may occur during the serves of athletes at different athletic levels may be important for the development of young tennis players.Therefore, I think the study is noteworthy.

Reviewer 3 ·

Basic reporting

The topic is not well presented, the text flow is not easy to follow and there are some evident structure, writing and conceptual inaccuracies. More than that, the text is not sufficiently based in up to date bibliography. This study has potential but not in the current form. As examples, it is possible to say that:
. the title is to long
. some expressions/sentences are are to understand (e.g. "high frame camera
technology", L59-61...)
. uncommon abbreviations should not be used
. speculative sentences should be avoided (e.g. "relatively little research has been conducted on the
biomechanics of the serve")
. some expressions are vague (e.g. "some variability", "in tennis technique", ...)
. some expressions/sentences should be supported by bibliographic references (e.g. L57-9, ...)
. In-text references are not always correctly displayed (e.g. L63).
. text font sizes are not consistent along the manuscript
. paragraphs sizes re not consistent along the manuscript
.

Experimental design

At the end of the second paragraph a study aim is mentioned. However, it is referred again in other place? Please review.
In the first sentence of the M&M the word "technical" is used. This term should be clarified along the manuscript.
The players recruitment should be more detailed by mentioning their level (using the levels referenced and described by the International Tennis Federation) and the exclusion and inclusion criteria should be more clear.
In the experimental approach to the problem, please display the type of the cameras and how was the calibration process? In L109 it was mentioned that the protocol set-up was changed to make the tests easier but it was not explained how this influenced the results.
The used warm-up and the time took to do two services of each type should be supported by proper bibliographic references.
Please be more clear regarding L115 (which is the best service?). Which are the criteria? It is also mentioned that the player performs two strong serves but is the aim of the second serve to be strongly performed? I understand that this is a criterion for the first serve, but not for the second.
An independent test was mentioned, but without displaying the significance level. It was also mentioned the correlation level, but without presenting any type of correlation test.
In L43, which were the criteria for defining the serve moments?

Validity of the findings

Numerical values should be included in the first sentences of the results section of the abstract.
In L148 the term "swing" was introduced without being mentioned before and without proper explanation.
From L153-7, methods and results are mixed and, in L154, variables are described but but without being presented in the methods section. In addition, the metrics are confusing.
The selected variables are scarce and more important ones might be used.
What exactly means the p<0.01 in L165? Please check the statistical analysis section to see if this information was included.
Then, differences between different groups are referred to, but no values for association between variables are displayed as stated in the statistical analysis section. The results section should be consistent with what was referred in the material and methods.
What does "Rocket" mean in the results related table?
The conclusion section is interesting but maybe some practical recommendations for coaches and players could be given so they can improve their training process regarding the studied topic.

Additional comments

In the Discussion section, the English language and text structure remains poor. These are some examples:
L186: this seems to be misplaced (better placed at the results section)
In L190 the expressions "brushing" and "dribble hits" are presented but without proper description. A deeper bibliographic review might be helpful to help better using concepts.
L191: is swing the same as speed?
From L192-6 the text is hard to follow. Please keep it simple and succinct.
It is missing a proper discussion of the values obtained and those from the literature.
In L207 authors refers to the conclusion of the study but we are not in that section
In L209 what do you mean by "shot positions"?
The discussion is very long compared to the introduction probably due to the redundant and extensive writing style.
Maybe the study limitations and recommendations for the future should be included.

---

## Round 0.2 · accepted · Accept

The authors addressed the reviewers comments. The manuscript is ready for publication.

Reviewer 2 ·

Basic reporting

Thank you to the authors for addressing recommendations. No comment

Experimental design

No commment

Validity of the findings

No comment

Additional comments

No commnent